# Role of Hydrogen Sulfide in Oncological and Non-Oncological Disorders and Its Regulation by Non-Coding RNAs: A Comprehensive Review

**DOI:** 10.3390/ncrna10010007

**Published:** 2024-01-18

**Authors:** Rana A. Youness, Danira Ashraf Habashy, Nour Khater, Kareem Elsayed, Alyaa Dawoud, Sousanna Hakim, Heba Nafea, Carole Bourquin, Reham M. Abdel-Kader, Mohamed Z. Gad

**Affiliations:** 1Biochemistry Department, Faculty of Pharmacy and Biotechnology, German University in Cairo (GUC), Cairo 11835, Egypt; 2Biology and Biochemistry Department, Faculty of Biotechnology, German International University (GIU), New Administrative Capital, Cairo 11835, Egypt; 3Pharmacology and Toxicology Department, Faculty of Pharmacy and Biotechnology, German University in Cairo (GUC), Cairo 11835, Egypt; 4Clinical Pharmacy Department, Faculty of Pharmacy and Biotechnology, German University in Cairo (GUC), Cairo 11835, Egypt; 5School of Pharmaceutical Sciences, Institute of Pharmaceutical Sciences of Western Switzerland, Department of Anaesthesiology, Pharmacology, Intensive Care and Emergency Medicine, University of Geneva, 1211 Geneva, Switzerland; carole.bourquin@unige.ch

**Keywords:** hydrogen sulfide, non-coding RNAs, CBS, CSE, 3-MST, cancer, non-oncological disorders

## Abstract

Recently, myriad studies have defined the versatile abilities of gasotransmitters and their synthesizing enzymes to play a “Maestro” role in orchestrating several oncological and non-oncological circuits and, thus, nominated them as possible therapeutic targets. Although a significant amount of work has been conducted on the role of nitric oxide (NO) and carbon monoxide (CO) and their inter-relationship in the field of oncology, research about hydrogen sulfide (H_2_S) remains in its infancy. Recently, non-coding RNAs (ncRNAs) have been reported to play a dominating role in the regulation of the endogenous machinery system of H_2_S in several pathological contexts. A growing list of microRNAs (miRNAs) and long non-coding RNAs (lncRNAs) are leading the way as upstream regulators for H_2_S biosynthesis in different mammalian cells during the development and progression of human diseases; therefore, their targeting can be of great therapeutic benefit. In the current review, the authors shed the light onto the biosynthetic pathways of H_2_S and their regulation by miRNAs and lncRNAs in various oncological and non-oncological disorders.

## 1. Introduction

Historically, hydrogen sulfide (H_2_S) has been a gas notorious for its noxious pungent odor of rotten eggs and for being identified as a toxic pollutant gas associated with several industries [1]. The identification of endogenous H_2_S in tissues has been reported in 1992, when Abe and Kimura reported on the endogenous production and signaling capacity of H_2_S as a neuromodulator [2].

The elucidation of its cellular signaling mechanisms led to the introduction of H_2_S into the family of gaseous signaling molecules known as gasotransmitters [3]. Until now, there have been three discovered gasotransmitters: nitric oxide (NO), identified in 1987, followed by discovery of carbon monoxide (CO) in 1990 and then the most recent member, H_2_S, in 1996 [4]. All the discovered gasotransmitters share a common set of characteristics, including regulated endogenous production, free permeation of cell membranes without transporters, and unique signaling function with certain interactions with cellular and molecular targets [3]. In addition, they have anti-inflammatory, anti-oxidant, anti-apoptotic, and cardioprotective activities. Yet, their point of difference is allocated in their endogenous production and their respective types of synthesizing enzymes, half-lives, and second messenger signaling [5].

H_2_S is colorless, flammable, and water-soluble gas with a characteristic smell of rotten eggs; it is now widely recognized as an endogenous biological mediator [6]. Similar to CO and NO, the old conventional thinking that all these gases are vicious and exert only detrimental influence on human health has gradually lost ground [7]. Since its discovery, H_2_S has been found to have vital functions in various physiological and pathological conditions [8,9,10,11]. Yet, its role in the malignant transformation process is still controversial and leads to a lot of puzzling conclusions. Moreover, unraveling H_2_S interactions within different tissues, its precise impact in different pathological conditions, and its interplay with other biochemical molecules and various signaling mediators is becoming even more complex [8]. It is worth noting that, recently, H_2_S has also been reported to be involved in the SARS-CoV-2 life cycle and associated symptoms [12], and others reported the protective role of H_2_S-releasing compounds in ameliorating SARS-CoV-2-associated lung endothelial barrier disruption [13].

Non-coding RNAs (ncRNAs) form a relatively recent class of post-transcriptional regulators that is widely expanding [14,15,16,17,18,19]. Recent work has redefined the perception of ncRNAs from ‘junk’ non-coding transcripts into functional regulatory mediators [20,21,22,23,24,25,26,27,28]. NcRNAs can regulate several cellular processes, such as chromatin re-modeling, post-transcriptional modifications, and the regulation of signal transduction pathways, through targeting several target mRNAs simultaneously [17,19,29,30]. Recently, we and others have highlighted the potential roles of ncRNAs in regulating H_2_S biosynthesis in several physiological and pathological contexts [31,32,33,34,35,36,37,38]. H_2_S and ncRNAs interact during the development and progression of several human diseases; therefore, their targeting can be of great therapeutic benefit [17]. It is also worth mentioning that miRNAs and lncRNAs are the dominating ncRNAs for regulating H_2_S biosynthesis in mammalian cells. However, this does not exclude the possibility that other ncRNAs are also involved in H_2_S biosynthesis regulation. We aimed, in this review, to shed the light onto the biosynthetic pathways of H_2_S and their regulation by ncRNAs in various oncological and non-oncological disorders.

## 2. Non-Coding RNAs: A Brief History

The emergence of small regulatory ncRNAs in the 1980s and 1990s completely shifted the pre-conceived notion of ncRNAs towards functional regulatory molecules [39,40,41,42]. Fire & Mello et al. indicated that double stranded RNAs (dsRNAs) could post-transcriptionally silence the complementary mRNAs in the nematode *Caenorhabditis elegans* via a process known as RNA interference [43,44]. Shortly afterwards, endogenous dsRNAs, such as small interfering RNAs (siRNAs) and microRNAs (miRNAs), were detected in a plethora of organisms, including plants, flies, and mammals [45]. The most thoroughly investigated classes of regulatory RNAs to date are piwi-associated RNAs, tRNAs, miRNAs, lncRNAs, and circRNAs (Figure 1) [45,46,47,48,49,50,51,52,53,54,55]. The non-coding portion of the mammalian transcriptome outweighs the protein-coding portion. This has been showcased in recently conducted studies, where only a mere 1.9% of the genome encodes for proteins [19,30,56,57,58]. In the coming sections of the review, the authors will focus on introducing the most well-studied ncRNAs.

### 2.1. MicroRNAs (miRNAs)

In 1993, lin-4 was the first miRNA discovered in *C. elegans* [42,59]. Lin-4 was found to target lin-4 mRNA [42,59]. However, its discovery did not attract much attention in the scientific community, as lin-4 was exclusively present in worms [60]. The second miRNA, let-7, was also discovered in *C. elegans*. In line with lin-4, let-7 plays a role in cell differentiation and proliferation. However, unlike lin-4, let-7 is specific to certain species, and it is evolutionarily conserved, highlighting its crucial roles in many organisms [61]. miRNAs are a class of single-stranded RNAs of 19–24 nucleotides (nt) [62,63]. More than 60% of mRNAs contain miRNA target sites in their 3′UTR region [29,64,65]. In addition, several miRNAs have shown themselves to be able to target an abundance of mRNAs, implying a complex and intricate mechanism of action for regulating mRNA [26,66]. Studies describing the implication of miRNAs in the development of several diseases [22,67,68], including cancers [20,67,69,70], cardiac diseases such as hypertrophy and ischemia [64,71,72], and mental disorders such as schizophrenia and major depression [73], were previously reviewed in one of our earlier publications [15].

### 2.2. Piwi-Interacting RNAs (piRNAs)

They are a group of small non-coding RNAs (sncRNAs) that was proved to exert defense against transposable elements (TEs) in germ lines [17,74]. Lately, ubiquitously expressed piRNAs have been reported in somatic and germ human cells, providing new insights about the diverse potential role of piRNAs [74]. piRNAs are composed of 26–31 nt [75]. It is worth mentioning that piRNAs could be classified into two sub-clusters according to their biogenesis: (1) the pachytene piRNA cluster, which is produced during meiosis and continuously expressed until the haploid spermatid stage, and (2) the pre-pachytene piRNA cluster, which mainly manifests in premeiotic germ cells [75]. Both gene clusters are transcribed into long primary RNAs that are further processed to create primary piRNAs, thereby acting as a guide for the genesis of secondary piRNAs [39]. Primary piRNAs biosynthesis machinery remains to be clearly elucidated; however, secondary piRNAs arise via an amplification mechanism known as the Ping-Pong amplification loop to augment piRNA sequences acting on active elements [76].

Concerning piRNAs function in mammals, piRNAs seem to primarily function in the germline, where they are thought to play a crucial role in repressing transposable and repetitive elements and diminishing their mobility in order to maintain the genomic integrity and stability of germline stem cells [53,77,78]. TEs are abundant repetitive DNA segments that have the capacity to replicate and insert themselves into new locations. This poses a huge threat to genomic stability, as the new insertions can cause DNA breaks and, consequently, the arrest of cell cycle, gene disruption, and higher chances of non-homologous recombination [79]. The extent of this looming threat and the criticality of piRNAs’ role were validated by Siomi et al., where their study showed that the absence of piwi family proteins in mice led to the inactivation of retrotransposons in the male germ line; consequently, this issue halted gametogenesis and resulted in male sterility [53].

### 2.3. Long ncRNAs (lncRNAs)

They are a class of diversified ncRNAs that are more than 200 nt in length and mainly found in the nucleus or cytoplasm, and as denoted by its name, it lacks protein-coding potential [80,81,82]. LncRNAs are grouped into five distinct categories: sense, antisense, bidirectional, intronic, and intergenic lncRNA [24,83]. Focusing more on their origin, lncRNAs can arise from varying sources, including [84] (a) the disruption of the translational reading frame of a protein-encoding gene (b), the reorganization of chromosomes, (c) the replication of a non-coding gene through retrotransposition, (d) non-encoding RNAs containing adjacent repeats, and (e) the insertion of transposable elements into a gene in such a fashion that they generate a functional, transcribed non-encoding RNA [14,18,23]. Despite the dissimilarity in their origin, studies have interestingly shown how lncRNAs share the same ultimate goal, i.e., to modulate gene expression [25,70,80,83].

### 2.4. Circular RNAs (circRNAs)

They are covalently closed circular RNAs that lack 5′-3′ polarity or a polyadenylated tail [45,85]. CircRNAs were first identified in 1976 [86] and have since been detected in humans, rats, mice, fungi, and other organisms [87,88,89,90,91,92]. Due to their structural specificity, unknown molecular role, and scarcity [93], circRNAs were initially thought to be ancient conserved molecules produced as splicing by-products and, hence, did not attract much attention at the time [16,94]. However, advances in the field of bioinformatics and high-throughput detection methods have led to the discovery of numerous circRNAs and helped us to gain more insight into their characteristics and functions [94]. It is worth noting that circRNAs are highly conserved molecules that are usually expressed in a tissue-specific and developmental stage-specific manner [95,96,97]. With respect to their function, circRNAs were found to act as miRNA sponges to deter mRNA translation [98,99] and induce gene expression by modulating splicing [100,101] or transcription, as well as interacting with RNA-binding proteins (RBPs) [102]. They are reportedly involved in multiple processes associated with disease progression, and, therefore, these molecules can pave the way for new potential opportunities and might serve as diagnostic and/or prognostic biomarkers [16,94].

## 3. Endogenous Hydrogen Sulfide (H_2_S) Biosynthesis

The endogenous generation of H_2_S mainly occurs through two main routes: enzymatic and non-enzymatic pathways. The enzymatic synthesis of H_2_S by mammalian tissues depends on two cytosolic pyridoxal-5′-phosphate-dependent enzymes: cystathionine β-synthase (CBS) and cystathionine γ-lyase (CSE), as well as a pyridoxal-5′-phosphate- independent mitochondrial enzyme known as 3-mercaptopyruvate sulfur-transferase (3-MST) [8]. CBS, CSE, and 3-MST are constitutive enzymes of varying expression levels in different tissues and organs; however, their levels are altered in various pathological conditions [103].

The expression level and activity of the aforementioned enzymes have been reported to be organ- and tissue-specific. For instance, CBS is mostly found in the brain, nervous system, and liver [2], while CSE is predominantly found in the liver and vascular system; however, 3-MST can be detected in the brain and vasculature [104]. It is also worth noting that 3-MST, located in the mitochondria, produces H_2_S together with cysteine aminotransferase (CAT) [105,106,107,108,109] (Figure 2). The literature shows that these enzymes coordinately regulate the trans-sulfuration activity and, thus, control physiological H_2_S levels in a complex and overlapping manner [8].

Apart from enzymatic synthetic pathways, endogenous H_2_S can also be produced through other non-enzymatic processes, as previously described [110]. To make it more clear, the non-enzymatic production of H_2_S occurs from either glucose via glycolysis or from phosphogluconate via NADPH oxidase [8]. H_2_S is also produced through the direct reduction of glutathione and elemental sulfur. The reduction of elemental sulfur to H_2_S is mediated by reducing equivalents of the glucose oxidation pathway like NADH or NADPH [111]. H_2_S formation from thiosulfate results from a reductive reaction involving pyruvate, which acts as a hydrogen donor. Thiosulfate is an intermediate of sulfur metabolism from cysteine and a metabolite of H_2_S that can also lead to the production of H_2_S [112].

## 4. H_2_S Enzymatic Machinery

### 4.1. Cystathionine-β-Synthase (CBS)

CBS is the most studied H_2_S-producing enzyme with regard structure and function. CBS gene is located on the long arm of chromosome 21 location 21q22.3. The crystalline structure of the active human CBS constitutes four 63-kDa subunits and is formed of 551 amino acids [113,114]. However, the shorter isozyme (≈48 kDa) was isolated from the liver and linked to increased catalytic activity in the enzyme [115]. Mechanistically, CBS is known to play an important role in amino acid metabolism. It facilitates the condensation of homocysteine with the L-serine amino acid into cystathionine, which is further broken down to form cysteine [116]. In addition to this canonical pathway, CBS has been found to synthesize H_2_S, depending on L-cysteine amino acid as a substrate and pyridoxal phosphate (PLP) as a coenzyme (Figure 2) [115]. CBS mainly resides in the cytosol; however, certain cell types have CBS located in the nucleus and mitochondria. The translocation of CBS from the cytosol to mitochondria can occur during states of hypoxia [117] or nucleolar stress [118]. As a H_2_S-synthesizing enzyme, CBS is implicated in numerous physiological events. Hence, many pathophysiological events have been linked to CBS dysregulation, including tumorigenesis [31,37,119].

### 4.2. Cystathionine-γ-Lyase (CSE)

CSE enzyme is encoded by the CTH gene on the short arm of chromosome 1 (1p31.1) and consists of 12 exons [120]. Another shorter transcription variant with lower activity was reported to be lacking exon 5 of the coding region [121]. CSE is a 405-amino acid polypeptide with a 45 kDa molecular weight. Similar to CBS, CSE catalytic activity is also PLP-dependent and depends on L-cysteine amino acid as a precursor for H_2_S synthesis [121]. However, unlike CBS, CSE is less investigated, and its regulation is less understood [122]. The pivotal role of CSE was markedly linked to the cardiovascular system [123]. In fact, endothelial cells and smooth-muscle cells in the blood vessels primarily synthesize H_2_S by CSE [122]. Yet, CSE dysregulation was also found to be associated with tumor progression. Amassing evidence shows that CSE plays crucial roles in numerous types of cancer cells. For instance, the inhibition or knockdown of CSE by shRNA halts the proliferation and migration of human colon cancer cells [124].

### 4.3. Mercaptopyruvate Sulfurtransferase (3-MST)

The human 3MST enzyme is encoded by the MPST gene on the long arm of chromosome 22 in the subtelomeric region q13.1 [125]. The MPST gene is 5.6 kbp in length with only two exon regions [126]. Unlike CBS and CSE, 3MST synthesizes H_2_S using 3-mercaptopyruvate (3-MP)—a product of L-cysteine metabolism by the cysteine aminotransferase (CAT) enzyme in the presence of α-ketoglutarate [127]. One more significant characteristic of 3MST is that it can be detected not only in the cytoplasm in a similar manner to CBS and CSE but also in the mitochondria [127]. Physiologically, 3MST plays a critical role in cellular metabolism and bioenergetics [128]. In recent years, there has been growing interest in the role of 3MST in cancer progression [129]. Some studies have shown that 3MST is upregulated in various types of cancers, as will be discussed later in this review. The expression levels of the three H_2_S-synthesizing enzymes in several types of cancer are displayed in Figure 3 and Table 1.

## 5. Role of H_2_S in Oncological Disorders and Its Regulation by ncRNAs

Most of the reported literature supports the pro-carcinogenic competence of endogenous H_2_S. Different oncogenic cascades have been drawn downstream for H_2_S, such as hastening of cell cycle progression, propagating anti-apoptotic signals and the induction of angiogenesis, thus supporting the novel concept of inhibiting H_2_S production as a promising strategy for cancer treatment [158].

### 5.1. Hastening of Cell Cycle

The cell cycle represents a series of highly regulated waves that control the transition of cells from the quiescent state (G_0_) to proliferative status and ensure the high conformity of the genetic transcript material [159]. Most adult mammalian cells are in the quiescent state, and their re-entry into the cell cycle requires the inactivation of the retinoblastoma protein (pRB) and the transcription of genes required for DNA replication and chromosome segregation [160]. Any disturbance in the regulation of cell cycle is a common feature in the development of human cancers [159]. A recent study reported that the knocking down of CSE in HCC cell lines induced G_1_/G_0_ phase arrest and decreased the cell population in S phase, while the cell population in S phase was induced via the administration of 100 µM of NaHS for 24 h [161]. NaHS is an inorganic sulfide salt that has been widely used as an H_2_S equivalent in many biological studies. Furthermore, NaHS has been widely used in studies investigating the role of exogenous H_2_S in cancer progression, as it rapidly releases H_2_S via simple hydrolysis under laboratory conditions [162]. Similarly, another study indicated that exogenous H_2_S (NaHS, 200–500 µM) served as a pro-proliferative factor that accelerates cell cycle progression in oral squamous cell carcinoma through the increased phosphorylation of Akt and extracellular signal-regulated kinase (ERK) [163]. Likewise, exogenous H_2_S picked up the pace of the cell cycle in HCT116 via another mechanism of action, where it induced the repression of the tumor suppressor protein (p21), one of the cyclin-dependent kinase (CDK) inhibitory proteins, and, thus, decreased the G_0_/G_1_ population and increased the cell population at the S phase [164].

### 5.2. Propagation of Anti-Apoptotic Signals

Apoptosis is a tightly nuanced cellular suicide program that is critical for the normal development and maintenance of tissue homeostasis in multi-cellular organisms. In cancer cells, apoptosis is suppressed, which is the main reason for the uncontrolled cellular proliferation and resistance to treatment [165]. The CSE/H_2_S pathway played a vital role in maintaining the cancerous behavior of hepatoma cells, and the inhibition of this pathway could significantly suppress the uncontrolled growth of hepatoma cells by stimulating mitochondrial apoptosis and suppressing cell growth signal transduction [166]. In the same way, another study reported that the treatment of PLC/PRF/5 hepatoma cells with 500 µM NaHS for 24 h markedly increased cellular viability and decreased the number of apoptotic cells [167]. In the same pattern, the exogenous application of 100–1000 µM NaHS for 10 min or the overexpression of CBS hindered apoptosis-induced 6-Hydroxydopamine in a human neuroblastoma cell line [168].

### 5.3. Induction of Angiogenesis

Angiogenesis, or the sprouting of new blood vessels from the existing vasculature, mediated by several growth factors, such as VEGF, EGF, IGF, and angiogenin [169], was reported to be a fundamental step in the process of metastasis [170]. A growing body of evidence highlights H_2_S as a vital stimulator of the malignant transformation process, starting from its support for the tumor cellular proliferation and ending with its invigoration of angiogenesis [171]. Recent studies indicate that H_2_S possesses pro-angiogenic effects mainly by increasing VEGF expression [172,173]. In colon and ovarian cancers, CBS knockdown reduces the metastatic potential of cancer cells and decreases the number of sprouted blood vessels, thus resulting in the attenuation of tumor growth [133,174]. Taken together, these results support the possibility of utilizing CBS, CSE and 3-MST as potential molecular targets for cancer treatment.

### 5.4. Colon Cancer

Also Szabo group highlighted that CBS is selectively upregulated in colon cancer tissues compared with non-malignant colonic epithelial tissues [130,174]. Through a series of studies involving the knocking in/out of CBS expression or CBS activity (allosteric activation via SAM or inhibition using amino-oxyacetic acid (AOAA) in the HCT116 colon cancer cell line, it was validated that CBS encourages tumor cell advancement [174]. Additionally, several human colon cancer cells displayed elevated CBS expression in comparison to normal non-malignant colon cancer cell lines (NCM356). In 2013, Szabo et al. reported that CBS silencing or inhibition resulted in the suppression of cell proliferation, as well as migration and invasion in HCT116 cells [174]. In line with these findings, SAM, an allosteric activator of CBS, was tested at low concentrations in HCT116, and it resulted in an increase in cell proliferation [175].

The induction of the Akt/PI3K signaling pathway is likely to trigger the pro-proliferative and migratory effects of CBS-synthesized H_2_S. These findings were validated in an in vivo model, where the growth rate in mice bearing xenografts of either HCT116 cells or patient-derived tumor tissue were significantly reduced following the silencing of CBS expression and/or inhibition of CBS expression via AOAA [176]. In addition, AOAA reduced the metastatic spread of HCT116 cells from the cecum to the liver in an orthotopic xenograft model of nude mice [177]. A recent study conducted by Guo et al. 2021 showed that the stable knockout of CBS using the CRISPR/Cas9 system in colon cancer cells caused the suppression of invasion and metastasis of cells. This study also showed that CBS knockdown reduced the expression of angiogenesis-related proteins such as VEGF [178].

Chen et al. studied the effects of miRNAs on the acquired resistance of 5-fluorouracil. The study revealed that the expression of miR-215-5p is altered following the knockdown of CBS, resulting in the increased sensitivity of HCT116 cells to 5-flurouracil. This effect was also noticed in an in vivo model [179]. Potential targets of miR-215-5p include epiregulin (EREG) and thymidylate synthetase (TYMS), whose expressions were attenuated via the inhibition of H_2_S synthesis, contributing to the reversal of acquired resistance to 5-flurouracil in HCT116 cell lines [179]. Regarding CSE, Olah et al. highlighted that in most colon cancer patients, CSE expression was quite low in both human biopsies. With respect to 3-MST, it displayed variable levels of expression. Nearly 50% of the individual sample pairs were of a higher expression level in the tumor compared to the normal counterparts, while the rest of the samples (50%) showed a low expression profile [131]. It is worth mentioning that upon performing literature screening, neither CSE nor 3MST regulation by ncRNAs in colon cancer had been previously investigated.

### 5.5. Breast Cancer

Several studies have shown that human breast cancer tissues and cells lines (MDA-MB-468, MCF-7, and Hs578T) reveal the marked overexpression of CBS, CSE, and 3MST compared to normal breast cells [31,132,180]. Recently, our research group has highlighted the potential regulation of CBS and CSE in breast cancer cell lines by miR-4317 [31] and miR-939-5p [181,182]. It is worth mentioning that, recently, miR-548 and miR-193 have been found by our research group to simultaneously target the three synthesizing enzymes in breast cancer: CBS, CSE and 3MST (Table 2) [132,183].

Digging deeper into the intricate crosstalk and potential interplays between H_2_S and different types of ncRNAs, our research group investigated the role of the lncRNA MALAT-1 in regulating STAT-3-regulated CSE in BC. Our results show that upon the knockdown of MALAT-1, a marked repression of STAT-3 and CSE was observed, thereby highlighting MALAT-1 as a novel upstream lncRNA regulating the STAT-3/CSE/ H2S axis in breast cancer [184]. Table 2 shows a summary of most of the ncRNAs (miRNAs or lncRNAs) regulating H_2_S-synthesizing enzymes in different pathological disorders.

### 5.6. Ovarian Cancer

Similar to results of colon and breast cancers, Bhattacharyya et al. have found that CBS is overexpressed in primary epithelial ovarian cancer human biopsies and cell lines, particularly in serous carcinoma, the most common histologic variant. The majority of ovarian cancer cells showed high CBS expression (both on the mRNA and protein levels) compared to normal ovarian surface epithelial cells [133]. In the same study, the authors also studied the regulatory role of CBS-derived H_2_S in modulating ovarian cancer cell proliferation, migration, and invasion in vitro and in vivo. In the in vitro model, a variety of genetically mediated stable silencing and pharmacological inhibition of CBS were used where either the silencing of CBS or CBS inactivation by AOAA was found to inhibit cellular viability and proliferation [133]. In the in vivo models, the silencing of CBS led to a significant decrease in tumor weight and the number of tumor nodules, while the inhibition of peritumor angiogenesis was confirmed to cause a marked reduction in CD31 staining [133]. It is worth mentioning that upon performing literature screening, the regulation of H_2_S-synthesizing enzymes by ncRNAs in ovarian cancer had not been previously investigated.

### 5.7. Melanoma

Penza et al. [134] evaluated the involvement of the H_2_S pathway in melanoma. An interesting study was performed involving more than 100 human samples that showed that elevated CSE expression increased from nevi to primary melanoma, decreased in tissue metastases, and was absent in lymph node metastases. On the other hand, the screening of the CBS expression level in the same cohort of patients revealed that CBS was absent in dysplastic nevi. Positive CBS expression was found in only 25% of the primary melanomas analyzed. Therefore, as opposed to other types of cancer reviewed in this review, CBS does not appear to play an dominant role in human melanoma. It is also worth noting that 3-MST expression showed a variable alteration in the analyzed human specimens (from nevi to metastasis) [134]. It is also worth noting that upon performing literature screening, H_2_S-synthesizing enzymes regulation by ncRNAs in melanoma had not been previously investigated.

### 5.8. Liver Cancer

Jia et al. [137] showed that CBS protein was highly expressed in the human hepatoma cell lines HepG2 and SMMC-7721. This observation paved the way for the idea that CBS/H_2_S is involved in HCC proliferation. This hypothesis was validated upon the silencing of the CBS using CBS siRNAs and the inactivation of CBS activity using AOAA, which evidently repressed the proliferation of HCC cell lines. Also, CBS siRNAs induced the apoptosis of SMMC-7721 cells, revealing the involvement of the CBS-induced H_2_S production in regulating hepatoma cell proliferation. In addition, an increase in the Bax/Bcl ratio and a significant upregulation of caspase-3 and PARP activities were evident following CBS knockdown [137].

### 5.9. Lung Cancer

Significantly higher protein levels of CBS, CSE, and 3-MST were detected in human lung adenocarcinoma samples compared to normal counterparts. Similarly, elevated expression of CBS, CSE, and 3-MST and H_2_S production were also reported in multiple lung adenocarcinoma cell lines (A549, H522 and H1944) compared to non-malignant lung epithelial cells (BEAS 2B) [138].

### 5.10. Bladder Cancer

The expression of H_2_S-producing enzymes in human bladder cancer tissues compared to their normal counterparts was investigated. Our results showed that H_2_S levels, as well as CBS, CSE, and 3-MST expression, were higher in bladder cancer biopsies compared to their normal counterparts. Notably, the expression of all three enzymes was correlated to different stages of bladder cancer [139]. Another study showed that the apoptosis of bladder cancer cells was heightened after the inhibition of H_2_S production by propargylglycine (PAG) and inhibited upon adding exogenous H_2_S. This was validated in in vitro, as well as in vivo, models. The stimulation of the Erk1/2 signaling pathway and the blockade of mitochondrial apoptosis were suggested as the probable mechanisms behind these results [185]. Exogenous H_2_S supplementation has also been shown to affect bladder cancer cells. It was found that preconditioning cells with sodium hydrosulfide (NaHS) boosted cells’ proliferation and invasion capacity. This was evident in the expression of matrix metalloproteinases (MMP) 2 and 9, which are crucial for the digestion of collagen IV and hydrolysis of extracellular matrix during invasion, as they increased in a dose-dependent manner after the treatment of bladder cancer cells with NaHS [186].

### 5.11. Leukemia

CBS was found to be elevated in bone marrow mononuclear cells isolated from newly diagnosed children with CML compared to age- and gender-matched control groups. Meanwhile, the CSE and 3-MST levels showed no significant differences between the two groups. At the in vitro level, in CML-derived K562 cells, the upregulation of CBS expression was demonstrated. CSE and 3-MST displayed no significant differences in their expression levels [140]. The silencing or pharmacological inhibition of CBS using AOAA induced a decrease in the cell proliferation of both K562 cells and bone marrow mononuclear cells from CML patients. This was mediated via the mitochondria-related apoptosis pathway [140].

### 5.12. Multiple Myeloma

The CBS expression level was significantly higher in patients with multiple myeloma than healthy donors [141]. The pharmacologic inhibition of H_2_S production using AOAA successfully inhibited the proliferation of the U266 myeloma cell line in a concentration-dependent manner. Exogenous NaHS has a pro-tumorigenic effect on U266 cells by enhancing cell cycle progression, which was diminished by using AOAA. AOAA treatment resulted in a decrease in the expression of anti-apoptotic Bcl-2 and an increase in caspase-3, thus promoting apoptosis [141].

**Table 2 ncrna-10-00007-t002:** Non-coding RNAs regulating H_2_S-synthesizing enzymes.

H_2_S Synthesizing Enzyme	miRNA	LncRNA	Cell Lines Investigated
CBS	miR-4317 [31] miR-6852 [187] miR-21 [32] miR-376a [188] miR-125b-5p [189] miR-203 [190] miR-215-5p [179] miR-939-5p [38] miR-548 [132] miR-193 [183]	LINC00336 [187] lncRNA CBSLR [191] LncRNA SNHG1 [188]	MDA-MB-231 MCF-7 A549 SPC-A-1 MGECs HCMIEC/D3 PC-12 SH-SY5Y HT-29 DLD-1 MKN45 MKN28
CSE	miR-4317 [31] miR-21 [32] miR-216a [192] miR-186 [193] miR-30a-5p [194] miR-328-3p [195] miR-30b-5p [196] miR-22 [197] miR-939-5p [38] miR-548 [132] miR-193 [183]	LncRNA Oprm1 [196] lncRNA MALAT-1 [184]	MDA-MB-231 MCF-7 HEK-293FT Primary cultured neonatal cardiomyocytes Mouse Airway Epithelial Cells (MAECs) THP-1 macrophages Vascular Smooth Muscle Cells (SMCs)
3-MST	miR-548 [132] miR-193 [183]	Not available	MDA-MB-231

## 6. Role of H_2_S in Non-Oncological Disorders and Its Possible Regulation by ncRNAs

### 6.1. Digestive System

It has been reported that the exogenous supplementation of H_2_S prevents ethanol-induced gastric damage in murine models and plays a shielding role against oxidative stress in rat gastric mucosal epithelium [198,199]. Takeuchi et al. reported that H_2_S is involved in the regulation of acid-induced bicarbonate secretion and mucosal protection in the duodenum [200]. H_2_S, along with other precursors comprising a dithiolethione moiety, are effective inducers of the antioxidant enzyme heme oxygenase-1 (HO-1). Notably, clinical evidence suggested that HO-1 promotes ulcer healing [201]. Other mechanisms thought to effectively contribute to the gastroprotective effect of H_2_S are increased epithelial secretion and mucosal blood flow; the reduction of leukocyte adhesion/infiltration; the downregulation of TNF-α, IL-1β and IFN-γ expression; and the scavenging of oxidative species [202]. H_2_S plays a potential role in resolving intestinal dysmotility in irritable bowel syndrome (IBS). A study conducted by Lin. et al. mentions the importance of CBS/H_2_S axis in stress-induced colonic hypermotility [203]. This is demonstrated using a stress-induced rat model of IBS, where CBS expression levels were significantly lower than the control and SAM administration groups, an activator of CBS [203].

Another study highlighted the importance of the H_2_S pathway in IBD, as the 3-MST expression level was significantly decreased in colonic tissues isolated from patients with ulcerative colitis or Crohn’s disease with prominent inflammatory responses. Low levels of 3-MST raised the levels of proinflammatory cytokines, ROS, and apoptosis. On the other hand, 3-MST overexpression alleviated these effects [204]. Additionally, 3-MST modulates apoptosis by increasing AKT expression and decreasing its phosphorylation.

### 6.2. Liver and Kidneys

The administration of H_2_S or H_2_S donor have been found to exert a protective role against ischemia-reperfusion damage in the liver and kidneys [205,206] and reduce the severity of liver injury [207]. In rats with ischemia-reperfusion injury, CSE and H_2_S levels are upregulated. It was found that the use of exogenous NaHS alleviated ischemia-reperfusion injury in liver hepatocytes in young rat models by activating Nrf2 pathway and reduction in miR-34a expression [208].

H_2_S has a vital nephroprotective role in diabetic kidney disease by alleviating renal glycative injury [206], inducing renal blood flow, increasing the glomerular filtration rate and urinary sodium excretion [209], and harnessing hyperhomocysteinemia-associated chronic renal failure [210]. Similarly, H_2_S was found to play a significant hepatoprotective role by diminishing stress-mediated liver injury and hepatic mitochondrial dysfunction in acutely ethanol-exposed mice [211], and it evidently relieves acetaminophen-induced hepatotoxicity in mice [212]. Likewise, H_2_S had protective role in liver fibrosis. There is evidence that CSE knockout increases cytokines, oxidative stress, inflammation, aggravating hepatitis, and liver fibrosis [213,214].

H_2_S also provided evidence for protection from cisplatin-induced acute kidney injury in dogs [215]. The administration of NaHS prior to cisplatin injection alleviated cisplatin-induced increase in the level of inflammatory mediators such as NF-κB, TNF-α, IL-1β in canine kidney tissues [215].

miR-21 was found to be upregulated in acute kidney injury (AKI) and may play a pathological role in the development of kidney disease associated with the H_2_S pathway [32]. For anti-miR-21, as well as exogenous H_2_S, GYY4137 reversed the changes in AKI through the enhancement of renal vascular density and blood flow in aged mice. Anti-miR-21 also induced the upregulation of CBS and CSE expression, as well as decreases in metalloproteinase-9 and collagen IV expression [32].

### 6.3. Diabetes Mellitus and Metabolism

Endogenous H_2_S acts as an antioxidant molecule protecting murine pancreatic β cells from oxidative stress and/or glucotoxicity-induced apoptosis. On the other hand, exogenous H_2_S through NaHS administration repressed reactive oxygen species (ROS) production by activating Akt signaling [216,217,218]. It seems that H_2_S biosynthesis decreases as the severity of the disease increases over time, and therapies based on the exogenous supplementation of H_2_S via administration of H_2_S donors might prove to be quite promising [219,220,221,222].

Similarly, the administration of NaHS to alleviate diabetic cardiomyopathy was recently studied in murine models [223]. Untreated diabetic models (type 2) showed decreased intracellular H_2_S levels, reduced CSE expression, altered cardiac function, and the interruption of calcium homeostasis, which activates the mitochondrial apoptosis pathway. Our results suggest that exogenous H_2_S improves myocardial systolic–diastolic function and reduces apoptosis in diabetic cardiomyopathy [223].

A study was conducted to understand the importance of dietary methionine restriction for improving metabolic health and boosting insulin sensitivity in mice models with high-fat-diet-induced obesity [195]. In this study, Wu et al. recommended that the reduction of dietary methionine enhances protein efficiency in mice and causes an increase in endogenous H_2_S production. An elevation in endogenous H_2_S is postulated to occur via miR-328-3p, which directly target CSE. This resulted in alleviating oxidative stress and ER stress, promoting protein homeostasis and metabolic efficiency in mice models with diet-induced obesity [195].

A common complication of diabetes mellitus is cardiovascular disorders, such as microcirculation dysfunction of the lower limbs associated with impaired angiogenesis. The impairment of angiogenesis is mainly caused by vascular endothelial dysfunction associated with hyperglycemia. H_2_S and miR-126-3p were recognized for their pro-angiogenesis effects [224]. The influence of H_2_S on miR-126-3p regulation was studied under high-glucose conditions. It was revealed that miR-126-3p expression is modulated by both diabetes and H_2_S. H_2_S upregulated miR-126-3p by inhibiting DNA methylation, as well as downregulating the DNMT1 (anti-angiogenic protein) level, which is primarily increased based on a high glucose level. In another context, CSE overexpression in human umbilical vein endothelial cells (HUVECs) has been shown to elevate miR-126-3p, thereby improving the function of endothelial cells and promoting angiogenesis [224]. In another study, Zhou Y et al. concluded that H_2_S promotes angiogenesis through the downregulation of the expression of miR-640 and by increasing the levels of HIF1α through the VEGFR2-mTOR pathway. Therefore, miR-640 plays a pivotal role in mediating the proangiogenic effect of H_2_S [225].

### 6.4. Cardiovascular System

Accumulated evidence highlights that H_2_S has potent effects on the myocardium and CSE overexpression is highly protective from I/R injury in the heart [226,227].

Exogenous H_2_S intake showed markedly positive outcomes in the pathological settings of atherosclerosis, myocardial I/R injury, chronic heart failure, and cardiopulmonary resuscitation in small animal models. The observed protective and improved outcomes are associated with enhanced cardiac mechanics, attenuated myocardial inflammation, Nrf2 activation, and reduced cardiomyocyte apoptosis [226,228,229,230,231].

MiR-30 family members are found to be upregulated and negatively correlated with the reported downregulation of CSE in the infarct and border zones following myocardial infarction (MI) in rat hearts. The dual-luciferase reporter assay confirmed that miR-30 directly targets CSE, and the fact that the inhibition of miR-30 can protect the MI heart by upregulating CSE was also confirmed in vivo [194] (Table 2).

As illustrated in the current review, it has come to our attention that miRNAs in particular are not only drawn as upstream regulators for the H_2_S-synthesizing enzymes but have also been reported to be drawn downstream for H_2_S, where either exogenously administered H_2_S or endogenously produced H_2_S affect an array of miRNAs as part of H_2_S-altered signaling cascades. For instance, exogenous H_2_S treatment upregulates cardioprotective miR-133a in primary cultures of neonatal rat cardiomyocytes and suppresses cardiomyocyte hypertrophy [232,233]. In addition, it attenuates myocardial ischemic and inflammatory injury via the induction of miR-21 [234]. Hu et al. investigated the effects of lncRNA Oprm1 on myocardial I/R injury and its underlying mechanism. Their findings revealed that lncRNA Oprm1 was significantly repressed in the I/R injury in an experimental animal model. Additionally, it exerts a stimulatory effect on CSE expression and H_2_S level. The dual-luciferase reporter gene assay unraveled the lncRNA/miRNA interplay between Oprm1 and miR-30b-5p that ultimately downregulates the expression of CSE, as shown in Figure 2. In addition, the results of the same study revealed that high levels of endogenous H_2_S together with the concomitant administration of Oprm1 in I/R injury can reduce apoptosis and protect the myocardium through the activation of the PI3K/Akt pathway and inhibition of HIF1-α activity [196].

Several experimental studies have reported that oxidant-induced leukocyte–endothelial cell interactions are mainly responsible for the microvascular dysfunction brought about via reperfusion [235,236]. In atherosclerosis, monocyte adherence to endothelial cells is stimulated by oxidized sulfur species [237]. Notably, H_2_S has wide-reaching anti-inflammatory and cytoprotective effects [105] and is an exceedingly potent inhibitor of leukocyte adherence to the vascular endothelium [238]. Thus, H_2_S might interfere in the inflammatory processes and reduce the tissue injury induced via neutrophils by stimulating apoptosis and/or the scavenging of neutrophil-derived hypochlorous acid (HOCl) [140]. Importantly, H_2_S is capable of downregulating numerous pro-inflammatory cytokines, including NF-κB, TNF-α, IL-1β, IL-6, and IL-8 [227,239,240,241], to modulate leukocyte adhesion and leukocyte-regulated inflammation, in order to mediate the cardio protection induced via ischemic postconditioning [242] and protect against NF-κB- and TNF-α-mediated endotoxic shock [243].

Moreover, a recent study demonstrated that treatment with GYY4137, a slow-releasing H_2_S donor, alleviated the endothelial dysfunction induced by TNF-α. The reduction of the activity pro-caspase 3, enhancement of mitochondrial function, and prevention of TNF-α-induced apoptosis are contributing factors involved in this improvement [244]. GYY4137 also upregulates miR-21 expression, which led to the activation of the Akt pathway [245].

It was reported that estrogen (E2) induces the transcription of CSE upon binding to the estrogen receptor α, which constitutively increases specificity protein-1 (SP1) activity. SP1 directly binds to the CSE promoter region to upregulate CSE transcription and H_2_S synthesis [197]. Moreover, the ectopic expression of miR-22 inhibits estrogen receptor α and, consequently, SP1 and CSE transcription [197]. This implies that miR-22 regulates H_2_S production in females, and consequently, this could be crucial in the treatment of females with cardiovascular disease [246].

Angiogenesis, the process of the formation of new capillaries from existing ones [247], plays a critical role in tumor growth and development [248,249]. A study by Cai et al. identified H_2_S as a promoter of angiogenesis [250]. Other studies have also reported the proangiogenic role of H_2_S; this was evident when NaHS demonstrated its capability for stimulating the proliferation and tube formation of endothelial cells (ECs), both in vitro and in vivo [173,250]. However, the signaling mechanisms behind it remain relatively elusive. miRNAs were suggested to be among the key players. The deletion of the RNA endonuclease Dicer, responsible for mediating miRNA maturation in the cytosol, and the subsequent aberrant vessel growth in mice models were the first pieces of experimental evidence that miRNAs have control over angiogenesis [251]. Despite establishing this correlation, many researchers are still intrigued with discovering miRNAs that specifically regulate H_2_S-mediated angiogenesis. Zhou et al. conducted a study highlighting the crosstalk between H_2_S and miR-640, as shown in Figure 4. This was evident upon the supplementation of vascular ECs with H_2_S, as miR-640 levels decreased. Alternately, inducing the expression of miR-640 inhibited the proangiogenic effect of H_2_S [225]. In addition, the knockdown of VEGFR2 or mTOR diminished the repression of miR-640, as well as the proangiogenic effect induced by H_2_S. It is worth noting that mTOR plays a significant role in modulating miR-640 [225]. Furthermore, miR-640 was found to suppress HIF1A expression by binding to the 3-UTR of its mRNA. It was also reported that HIF1A suppression could be overturned by treating the cells with H_2_S [225].

### 6.5. Nervous System

H_2_S is avital neuromodulator and neuroprotective agent in the central nervous system [252]. Guo et al. highlighted that H_2_S can protect the brain from I/R injury by maintaining mitochondrial function, repressing pro-inflammatory factors, attenuating ROS, and hampering apoptosis [253]. Li et al. indicated that H_2_S protected the spinal cord and stimulated autophagy via miR-30c in a rat model of spinal cord I/R injury [254]. Although accumulating evidence crystallized the importance of H_2_S as a neuroprotectant, the underlying mechanisms of action of this gasotransmitter in spinal cord ischemia-reperfusion injury (SCII) were still indistinct [255]. This prompted Liu et al. to investigate whether cancer susceptibility candidate 7 (CASC7), a dual-localized lncRNA, was implicated in the neuroprotective effects of H_2_S in SCII. Their results have shown that in SCII rat, CasC7 expression was significantly reduced, while miR-30c expression levels were increased. In contrast to these observations, upon NaSH (H_2_S donor) preprocessing, CasC7 expression was elevated, while that of miR-30c was decreased [255]. These results proposed that preconditioning with NaSH reduced the spinal cord infarct zone and reversed the CasC7 and mir-30c usual expression pattern. Therefore, this study unveiled the regulatory role exerted by H_2_S in modulating the expression of CasC7 and miR-30c in SCII rat models, as well as validated the neuroprotective role of hydrogen sulfide in the spinal cord [255].

Emerging experimental data suggest that H_2_S might carry therapeutic potential for Alzheimer’s patients, since it reduces the mRNA and protein levels of β-amyloid precursor protein-cleaving enzyme 1 [256,257].

Another study demonstrated the neuroprotective role of H_2_S in hypoxic injury by upregulating the expression of heat shock protein (HSP) 90, a ubiquitous molecule that is associated with cell survival [258,259]. Osborne et al. reported that the slow-releasing H_2_S prodrug ACS67 diminishes retinal ischemic damage following the elevation of retinal pressure in rats [260]. The authors of this study pointed out that the neuroprotective effect of ACS67 possibly involves the stimulation of GSH formation among several other mechanisms. Biermann et al. also recently reported that H_2_S preconditioning visibly diminishes retinal ganglion cells (RGCs) apoptosis following retinal ischemia-reperfusion injury [261]. In addition, Mikami et al. displayed the ability of H_2_S to protect the retina from light-induced damage via the regulation of intracellular calcium by activating vacuolar type H+-ATPase [262]. It has been found that hyperhomocysteinemia can most commonly cause mental illness, seizures, and Alzheimer’s disease through Hcy-induced oxidative stress. Tyagi et al. nominated H_2_S as a form of shield for the brain that could potentially be a promising therapeutic candidate for the treatment of hyperhomocysteinemia-associated pathologies, such as stroke and neurologic disorders [263].

Chen Z et al. identified the miR-485-5p/TRADD axis as a key player in H_2_S protection against TNF-α-induced neuronal cell apoptosis. Using a luciferase-reporting gene assay, Western blot, and qRT-PCR, Chen Z et al. confirmed that TRADD is a direct target for miR-485-5p, while the miR-485-5p/TRADD axis can be drawn downstream for H_2_S in neuronal cells [264].

Liu Y et al. shows that H_2_S significantly attenuated the increase in Rho-associated protein kinase 2 (ROCK2) and the decrease in miR-135a-5p by MPTP (1-methyl-4-phenyl-1,2,3,6-tetrahydropyridine) in neuronal cells [265].

MiR-125b-5p has been identified as the culprit in an array of neurological disorders, such as Parkinson’s disease [265], Alzheimer’s disease (AD) [266], and pregnancy-related complications [267]. However, few studies have attempted to study its role in ischemic stroke, especially the interaction with H_2_S in cerebral hypoxia-induced injury [189]. In a study by Shen et al., the relationship between miR-125b-5p and H_2_S production, as shown in Figure 4, as well as the regulatory mechanism exerted by miR-125b-5p on H_2_S generation in oxygen/glucose deprivation (OGD)-induced injury in neuron-like rat pheochromocytoma (PC-12), was investigated. The overexpression of miR-125b-5p reduced CBS expression and H_2_S generation, which consequently exacerbated OGD injury. In contrast, the downregulation of miR-125b-5p increased CBS expression and H_2_S generation and prohibited PC-12 cells from OGD injury. These results suggested that miR-125b-5p might participate in cerebral hypoxia injury by modulating H_2_S levels [189].

## 7. Conclusions and Future Recommendations

In conclusion, the exploration of H_2_S and its synthesizing enzymes has revealed a multifaceted landscape of cellular regulation. The intricate balance of H_2_S production, facilitated by CBS, CSE, and 3-MST, emerges as a critical factor in coordinating various physiological and pathological processes. This review demonstrates the comprehensive stratification of the expression pattern of those enzymes in both oncological and non-oncological contexts and the pivotal role of ncRNAs in orchestrating the web of interactions between H_2_S and its synthesizing enzymes.

In essence, the convergence of research on H_2_S, its synthesizing enzymes, and the regulatory role of ncRNAs signifies a paradigm shift in our comprehension of cellular signaling. As we continue to unravel the complexities of these interactions, the potential for groundbreaking advancements in therapeutic strategies targeting H_2_S-associated pathways becomes increasingly promising, offering hope for developing more effective and tailored treatments in the realm of numerous disorders.

It is also worth noting that until now, miRNAs and lncRNAs have been the main candidates from the ncRNA family reported to modulate the H_2_S biosynthesis under different physiological and pathological conditions. However, other classes of ncRNAs such as circRNAs and piwi RNAs should be evaluated in terms of regulating H_2_S and its synthesizing enzymes in different pathological conditions. Nonetheless, crosstalk between different classes of ncRNAs should also be probed to explore ceRNA circuits that could be regulating the H_2_S-synthesizing enzymes in several oncological and non-oncological contexts. On a therapeutic level, understanding the roles of miRNAs and lncRNAs in H_2_S-regulated pathways may lead to the identification of novel therapeutic targets for all highlighted diseases associated with H_2_S dysregulation in this review. Future research could explore the development of the RNA-based therapeutics or small molecules targeting these ncRNAs for disease intervention.

## Figures and Tables

**Figure 1 ncrna-10-00007-f001:**
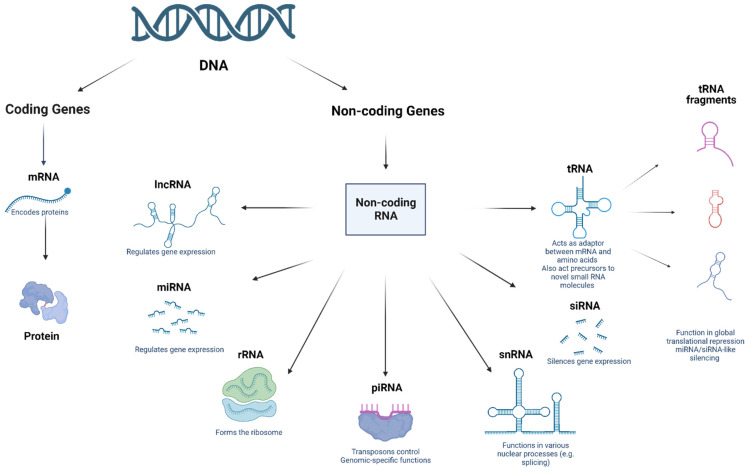
Different types of non-coding RNAs.

**Figure 2 ncrna-10-00007-f002:**
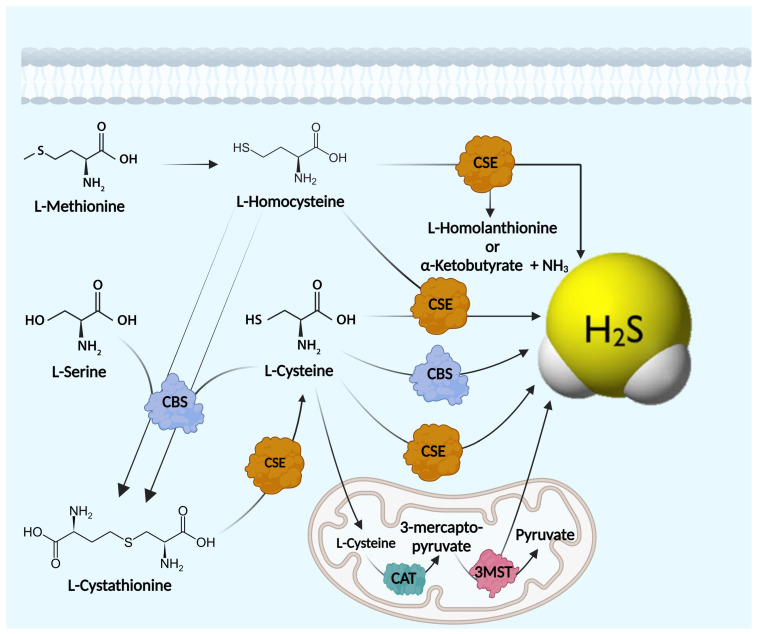
Hydrogen sulfide biosynthesis.

**Figure 3 ncrna-10-00007-f003:**
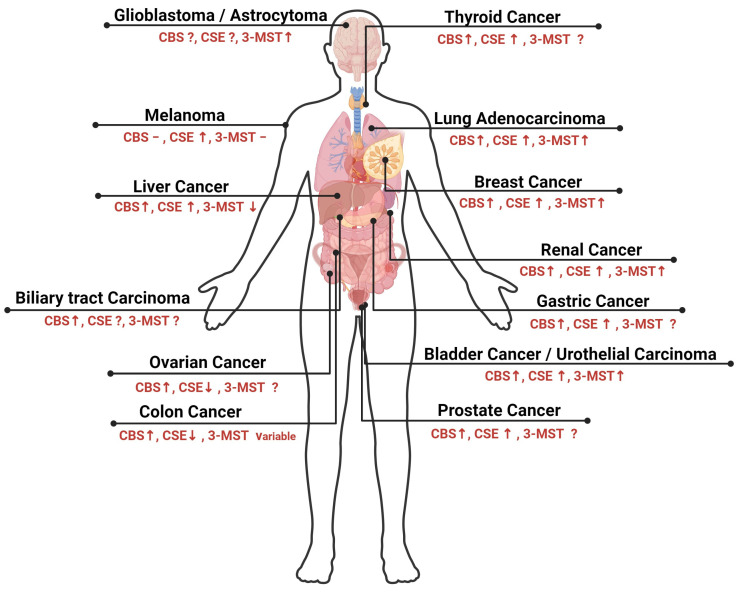
Differential expression of H_2_S-synthesizing enzymes in solid malignancies.

**Figure 4 ncrna-10-00007-f004:**
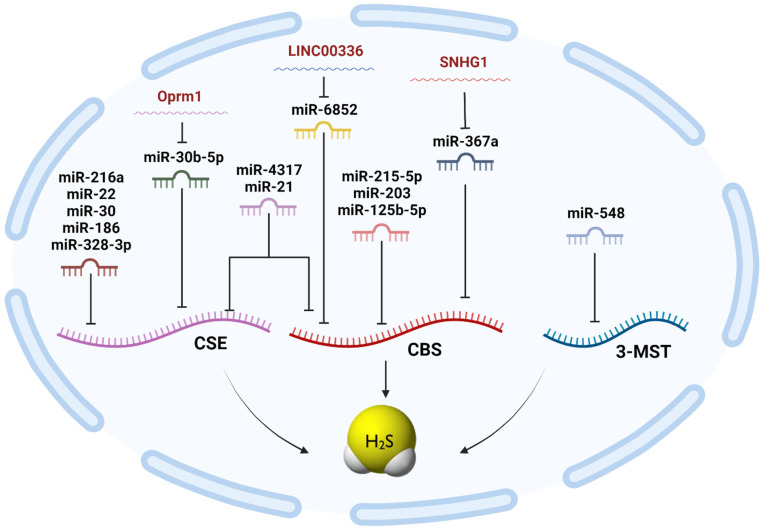
A snapshot of different classes of non-coding RNAs regulating H_2_S machinery.

**Table 1 ncrna-10-00007-t001:** Expression levels of H_2_S-synthesizing enzymes in several oncological phenotypes.

Type of Cancer	Expression Level of CBS	Expression Level of CSE	Expression Level of 3-MST	Cancer Cell Lines/Cancer Tissues	References
Colon cancer	High	Low	Variable	Human tissues HCT116 HT29	[130,131]
Breast cancer	High	High	High	Human tissues	[31,132]
Ovarian cancer	High	Low	Not investigated	OV167 OV202 SKOV3 A2780	[133]
Melanoma	Not altered	High	Not altered	A375 Sk-Mel-5 Sk-Mel-28 PES 43	[134]
Liver cancer	High	High	Low	HepG2 SMMC-7721 QGY-7703 BEL-7402 BEL-7404 MHCC-LM3 Huh7 Hep3B	[135,136,137]
Lung adenocarcinoma	High	High	High	A549 H522 H1944	[138]
Bladder cancer	High	High	High	Human tissues	[139]
Leukemia	High	Not altered	Not altered	CML-derived K562 cells	[140]
Multiple Myeloma	High	Not investigated	Not investigated	Human tissues	[141]
Prostate cancer	High	High	Not investigated	LNCap PC3 LNCaP-B	[142,143,144]
Gastric cancer	High	High	Not investigated	SGC-7901 AGS	[145,146]
Renal cancer	High	High	High	RCC4	[147,148]
Glioblastoma	Not investigated	Not investigated	High	C6, U87	[37,149,150]
Thyroid cancer	High	High	Not investigated	TPC1 TT ARO	[151,152]
Urothelial carcinoma	High	High	High	5637 UM-UC-3 EJ	[129,139,147,153]
Biliary tract carcinoma	High	Not investigated	Not investigated	GB-H3 GB-D1 TFK-1 HUCCT-1	[154]
Astrocytoma	Not investigated	Not investigated	High	U373	[150]
Neuroblastoma	Not investigated	Not investigated	High	SH-SY5Y	[155,156]
Oral squamous cell carcinoma	High	High	High	OSCC	[157]

## Data Availability

Not applicable.

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
