# Peer review of "Role of Hydrogen Sulfide in Oncological and Non-Oncological Disorders and Its Regulation by Non-Coding RNAs: A Comprehensive Review"

_ncrna, 2024, doi:10.3390/ncrna10010007_

Round 1

Reviewer 1 Report

Comments and Suggestions for Authors

The review is thorough and encompasses a broad range of relevant literature, providing a substantial overview and offering insights that are likely to be valuable to researchers in this area. However, the paper currently lacks a final conclusion and outlook section. I recommend the authors to add a concluding section that summarizes the main findings and their implications. Additionally, an outlook for future research in this area would greatly enhance the paper's utility, providing guidance for subsequent studies and developments in the field. In addition, there are a few typographical errors in the manuscript. A careful revision is suggested to identify and fix these errors. Given the strengths of the paper and the relatively minor nature of the required revisions, my recommendation is to Accept after Minor Revisions.

Author Response

Kindly find herein a revised version of our manuscript, entitled “Role of Hydrogen Sulfide in Oncological and Non-Oncological Disorders and its Regulation by Non-coding RNAs: A Comprehensive Review”. Manuscript ID is ncRNA-2778534.

We are gratified for the opportunity to submit a revised version of our manuscript to your journal. In reply to the valuable comments of the reviewers and in the provision of the revised manuscript, I am sending you this letter providing the detailed changes that have been entailed in the revised manuscript (in red) for your kind consideration.

Dear Reviewers,

We are thankful for your thorough review of our manuscript that aims for the improvement and scientific enrichment of our paper. All the issues noted by reviewers have been taken into consideration and extensively revised. The comments and inquiries are shown in Bold and the replies are shown in red.

Reviewer 1:

The review is thorough and encompasses a broad range of relevant literature, providing a substantial overview and offering insights that are likely to be valuable to researchers in this area. However, the paper currently lacks a conclusion and outlook section. I recommend the authors to add a concluding section that summarizes the main findings and their implications. Additionally, an outlook for future research in this area would greatly enhance the paper's utility, providing guidance for subsequent studies and developments in the field. In addition, there are a few typographical errors in the manuscript. A careful revision is suggested to identify and fix these errors. Given the strengths of the paper and the relatively minor nature of the required revisions, my recommendation is to Accept after Minor Revisions.

We would like to thank the reviewer for his/her positive feedback and his/her recommendations. A conclusion and future recommendation sections have been employed in the revised attached manuscript pages 27, lines (618-637).

Thank you very much

Hope the revised manuscript finds your acceptance

Best Regards,

Rana Ahmed Youness, PhD

Assistant Professor of Molecular Biology and Biochemistry and Group Leader, Faculty of Biotechnology, German International University (GIU), New Administrative Capital, 11586, Cairo, Egypt. E-mail: [email protected]; [email protected]

Reviewer 2 Report

Comments and Suggestions for Authors

The manuscript provides a resource for people inside and outside of the field  of H2S biosynthesis, and it shows important ncRNA interactions with H2S biosynthesis. The paper is well organized and easy to follow. I only have corrections and suggestions for improving the text. 

1.    Abstract

Needs improving:

Consider changing to: “A revolution in the field of gasotransmitters has evolved since the discovery in1992 of the endogenous production of widely known notorious gas, hydrogen sulfide (H2S) in mammalian tissues. Since then, H2S has been casted as an endogenous modulator of several a large number of cellular and molecular events in physiological and pathological conditions.”

Leave out your own work in the Abstract as this is a comprehensive review. Your work is mentioned in the Introduction and perhaps elaborate more about your studies in this section. 

The association of numerous ncRNAs in H2S biosynthesis is very striking (Table 2 and Fig.4). If this is highlighted in the Abstract it should generate considerable interest with ncRNA readers as you show that there are a large number of miRNAs and several lncRNA associated with the H2S biosynthetic pathway and disease conditions. A strong statement on this would be useful. 

Define what you mean by “ncRNAs interact in several directions”

2.    Text  

P.3 Fig 1

To update the ncRNA summary, add a section on tRNA ncRNA fragments and their involvement in regulation and add references:

tRNA-Derived Fragments (tRFs): Emerging New Roles for an Ancient RNA in the Regulation of Gene Expression.

Keam SP, Hutvagner G.Life (Basel). 2015 Nov 27;5(4):1638-51. doi: 10.3390/life5041638.

Emerging roles of tRNA-derived fragments in cancer.

Fu M, Gu J, Wang M, Zhang J, Chen Y, Jiang P, Zhu T, Zhang X.Mol Cancer. 2023 Feb 13;22(1):30. doi: 10.1186/s12943-023-01739-5.

The Typical tRNA Co-Expresses Multiple 5' tRNA Halves Whose Sequences and Abundances Depend on Isodecoder and Isoacceptor and Change with Tissue Type, Cell Type, and Disease.

Akins RB, Ostberg K, Cherlin T, Tsiouplis NJ, Loher P, Rigoutsos I.Noncoding RNA. 2023 Nov 6;9(6):69. doi: 10.3390/ncrna9060069.

In Fig 1, I suggest putting an arrow to the right of the tRNA drawing, add a diagram of tRNA fragments (similar to Fig. 1 in paper: Emerging roles of tRNA-derived fragments in cancer Min Fu et al, and with functions shown underneath, e.g. regulation of gene expression, etc. which could follow the format of the other ncRNAs that you show.

p. 2 Please correct the history:

“The emergence of small regulatory ncRNAs in the 1990s completely shifted the pre- conceived notion of ncRNAs towards functional regulatory molecules [31].” Correct this statement. The first regulatory ncRNA that controls gene expression, micF RNA, was discovered in the 1980s, six years before the miRNA work.

Consider changing to:

The emergence of small regulatory ncRNAs in the 1980s and 1990s completely shifted the pre- conceived notion of ncRNAs towards functional regulatory molecules [1-4]. 

Add references:

1 The isolation and characterization of RNA coded by the micF gene in Escherichia coli.; Andersen J, Delihas N, Ikenaka K, Green PJ, Pines O, Ilercil O, Inouye M. Nucleic Acids Res. 1987 Mar 11;15(5):2089-101. doi: 10.1093/nar/15.5.2089.\

2  The function of micF RNA. micF RNA is a major factor in the thermal regulation of OmpF protein in Escherichia coli.

Andersen J, Forst SA, Zhao K, Inouye M, Delihas N.J Biol Chem. 1989 Oct 25;264(30):17961-70.

3. The C. elegans heterochronic gene lin-4 encodes small RNAs with antisense complementarity to lin-14.Lee RC, Feinbaum RL, Ambros V.Cell. 1993 Dec 3;75(5):843-54. doi: 10.1016/0092-8674(93)90529-y

4  Discovery and characterization of the first non-coding RNA that regulates gene expression, micF RNA: A historical perspective. Delihas N. World J Biol Chem. 2015 Nov 26;6(4):272-80. doi: 10.4331/wjbc.v6.i4.272.).

At the end of the section “2. Non-coding RNAs: a brief history”: perhaps state something like, so far, only miRNAs and lncRNAs have been shown to have an involvement in H2S biosynthesis, but the possibility that other ncRNAs also be involved should not be excluded. 

p. 13, paragraph 4. Define what you mean by drawn upstream:

“It has been reported that miRNAs are not drawn upstream the H2S synthesizing enzymes regulating the biosynthesis of H2S,” 

 MiR-640 is discussed in p.13 and p.15 but it is not shown in Table 2 or Fig 4. Please correct.

3.    p.16, Add Conclusions

Summarize and highlight significance of ncRNA in H2S regulation.

Add possible future directions in H2S research. 

Author Response

Kindly find herein a revised version of our manuscript, entitled “Role of Hydrogen Sulfide in Oncological and Non-Oncological Disorders and its Regulation by Non-coding RNAs: A Comprehensive Review”. Manuscript ID is ncRNA-2778534.

We are gratified for the opportunity to submit a revised version of our manuscript to your journal. In reply to the valuable comments of the reviewers and in the provision of the revised manuscript, I am sending you this letter providing the detailed changes that have been entailed in the revised manuscript (in red) for your kind consideration.

Dear Reviewers,

We are thankful for your thorough review of our manuscript that aims for the improvement and scientific enrichment of our paper. All the issues noted by reviewers have been taken into consideration and extensively revised. The comments and inquiries are shown in Bold and the replies are shown in red.

Reviewer 2:

1.    Abstract

Needs improving:

Consider changing to: “A revolution in the field of gasotransmitters has evolved since the discovery in1992 of the endogenous production of widely known notorious gas, hydrogen sulfide (H2S) in mammalian tissues. Since then, H2S has been casted as an endogenous modulator of several a large number of cellular and molecular events in physiological and pathological conditions.”

According to the reviewer’s recommendation, these sentences have been changed in the revised manuscript and the abstract has been re-structured after considering all reviewer’s comments and considerations page 2, lines (40-52).

Leave out your own work in the Abstract as this is a comprehensive review. Your work is mentioned in the Introduction and perhaps elaborate more about your studies in this section.

According to the reviewer’s recommendation, our work has been removed from the abstract and re-structuring of the abstract section has been performed based on your recommendations.

The association of numerous ncRNAs in H2S biosynthesis is very striking (Table 2 and Fig.4). If this is highlighted in the Abstract it should generate considerable interest with ncRNA readers as you show that there are a large number of miRNAs and several lncRNA associated with the H2S biosynthetic pathway and disease conditions. A strong statement on this would be useful.
According to the reviewer’s recommendation, these striking results related to ncRNAs association with H2S synthesizing enzymes have been added in the abstract section of the revised manuscript page 2, lines (47-51).

Define what you mean by “ncRNAs interact in several directions”

In reply to the reviewer’s recommendation, this sentence has been replaced by a more elaborative and descriptive sentence highlighting the intricate crosstalk between different classes of ncRNAs

  1.   Text 

    P.3 Fig 1
    To update the ncRNA summary, add a section on tRNA ncRNA fragments and their involvement in regulation and add references:
    tRNA-Derived Fragments (tRFs): Emerging New Roles for an Ancient RNA in the Regulation of Gene Expression.
    Keam SP, Hutvagner G.Life (Basel). 2015 Nov 27;5(4):1638-51. doi: 10.3390/life5041638.
    Emerging roles of tRNA-derived fragments in cancer.
    Fu M, Gu J, Wang M, Zhang J, Chen Y, Jiang P, Zhu T, Zhang X.Mol Cancer. 2023 Feb 13;22(1):30. doi: 10.1186/s12943-023-01739-5.
    The Typical tRNA Co-Expresses Multiple 5' tRNA Halves Whose Sequences and Abundances Depend on Isodecoder and Isoacceptor and Change with Tissue Type, Cell Type, and Disease.
    Akins RB, Ostberg K, Cherlin T, Tsiouplis NJ, Loher P, Rigoutsos I.Noncoding RNA. 2023 Nov 6;9(6):69. doi: 10.3390/ncrna9060069.
    According to the reviewer’s recommendation, the summary section has been updated and all recommended references have been added to the revised manuscript, Page 4, line 99 and references (52-53).

In Fig 1, I suggest putting an arrow to the right of the tRNA drawing, add a diagram of tRNA fragments (similar to Fig. 1 in paper: Emerging roles of tRNA-derived fragments in cancer Min Fu et al, and with functions shown underneath, e.g. regulation of gene expression, etc. which could follow the format of the other ncRNAs that you show.
According to the reviewer’s recommendation, A revised version of Figure 1 has been uploaded with the revised manuscript

  1. 2 Please correct the history:
    “The emergence of small regulatory ncRNAs in the 1990s completely shifted the pre- conceived notion of ncRNAs towards functional regulatory molecules [31].” Correct this statement. The first regulatory ncRNA that controls gene expression, micF RNA, was discovered in the 1980s, six years before the miRNA work.

Consider changing to:
The emergence of small regulatory ncRNAs in the 1980s and 1990s completely shifted the pre- conceived notion of ncRNAs towards functional regulatory molecules [1-4].
Add references:
1 The isolation and characterization of RNA coded by the micF gene in Escherichia coli.; Andersen J, Delihas N, Ikenaka K, Green PJ, Pines O, Ilercil O, Inouye M. Nucleic Acids Res. 1987 Mar 11;15(5):2089-101. doi: 10.1093/nar/15.5.2089.\
2  The function of micF RNA. micF RNA is a major factor in the thermal regulation of OmpF protein in Escherichia coli.
Andersen J, Forst SA, Zhao K, Inouye M, Delihas N.J Biol Chem. 1989 Oct 25;264(30):17961-70.
3. The C. elegans heterochronic gene lin-4 encodes small RNAs with antisense complementarity to lin-14.Lee RC, Feinbaum RL, Ambros V.Cell. 1993 Dec 3;75(5):843-54. doi: 10.1016/0092-8674(93)90529-y
4  Discovery and characterization of the first non-coding RNA that regulates gene expression, micF RNA: A historical perspective. Delihas N. World J Biol Chem. 2015 Nov 26;6(4):272-80. doi: 10.4331/wjbc.v6.i4.272.).
We are extremely grateful for the meticulous comment from the reviewer, this correction has been done in the revised manuscript and all recommended references have been updated in the revised manuscript as per your recommendations. This is now in the revised manuscript page 3, lines (91-93) and references (38-40).

At the end of the section “2. Non-coding RNAs: a brief history”: perhaps state something like, so far, only miRNAs and lncRNAs have been shown to have an involvement in H2S biosynthesis, but the possibility that other ncRNAs also be involved should not be excluded.
According to the reviewer’s recommendation, this sentence has been added in the revised manuscript page 3, lines (85-88).
p. 13, paragraph 4. Define what you mean by drawn upstream: “It has been reported that miRNAs are not drawn upstream the H2S synthesizing enzymes regulating the biosynthesis of H2S,”
In reply to the reviewer’s comment, this sentence has been rephrased to be more understandable and scientifically elaborated in the revised manuscript

 MiR-640 is discussed in p.13 and p.15 but it is not shown in Table 2 or Fig 4. Please correct.
We are extremely grateful for the meticulous comment from the reviewer, Table 2 and Figure 4 are mainly dedicated to miRNAs and lncRNAs that are experimentally validated to directly target any of the H2S synthesizing enzymes. However, miR-640, miR-34a and miR-245-5p mentioned in the review are miRNAs that are reported to be altered upon exogenous H2S administration but are not validated regulators for H2S synthesizing enzymes. However, all validated ncRNAs that are directly targeting either CBS or CSE or 3-MST are thoroughly revised and very recent miRNAs from literature were included in text, table and figures such as miR-193 that was recently reported to target 3-MST.

  1.   p.16, Add Conclusions
    Summarize and highlight significance of ncRNA in H2S regulation.
    Add possible future directions in H2S research.
    According to the reviewer’s recommendation, a conclusion and future recommendation sections have been added in the revised manuscript attached Pages 26-27, lines (618-637)..

Thank you very much

Hope the revised manuscript finds your acceptance

Best Regards,

Rana Ahmed Youness, PhD

Assistant Professor of Molecular Biology and Biochemistry and Group Leader, Faculty of Biotechnology, German International University (GIU), New Administrative Capital, 11586, Cairo, Egypt. E-mail: [email protected]; [email protected]

Reviewer 3 Report

Comments and Suggestions for Authors

This is a very due and quite comprehensive paper summarizing numerous data accumulated in recent decades concerning the interplay between ncRNAs, synthesis of H2S, and various forms of cancer in humans. The authors described in detail the transsulfuration pathways of H2S production but did not mention other "chemical" ways of H2S synthesis without involving PBS, CSE, and MST. When describing various processes and pathologies where H2S is involved it is necessary to mention the involvement of H2S in the regulation of acute respiratory distress syndrome induced by RNA viruses including SARS. 

Comments on the Quality of English Language

The language is plausible but it would be nice to ask a native speaker to read the ms.

Author Response

Kindly find herein a revised version of our manuscript, entitled “Role of Hydrogen Sulfide in Oncological and Non-Oncological Disorders and its Regulation by Non-coding RNAs: A Comprehensive Review”. Manuscript ID is ncRNA-2778534.

We are gratified for the opportunity to submit a revised version of our manuscript to your journal. In reply to the valuable comments of the reviewers and in the provision of the revised manuscript, I am sending you this letter providing the detailed changes that have been entailed in the revised manuscript (in red) for your kind consideration.

Dear Reviewers,

We are thankful for your thorough review of our manuscript that aims for the improvement and scientific enrichment of our paper. All the issues noted by reviewers have been taken into consideration and extensively revised. The comments and inquiries are shown in Bold and the replies are shown in red.

Reviewer 4:
This is a very due and quite comprehensive paper summarizing numerous data accumulated in recent decades concerning the interplay between ncRNAs, synthesis of H2S, and various forms of cancer in humans. The authors described in detail the transsulfuration pathways of H2S production but did not mention other "chemical" ways of H2S synthesis without involving PBS, CSE, and MST.

We would like to thank the review for his/her positive feedback and comments. According to the reviewer’s recommendation, all enzymatic and non-enzymatic biochemical synthetic pathways for H2S have been added in the revised manuscript Pages 6-7, lines (169-174). As well as a new section highlighting the mechanism of action of H2S in different physiological and pathological conditions have been added to the manuscript, Pages 12-13, lines (243-292).

When describing various processes and pathologies where H2S is involved it is necessary to mention the involvement of H2S in the regulation of acute respiratory distress syndrome induced by RNA viruses including SARS.

According to the reviewer’s comment, the involvement of H2S in the regulation of acute respiratory distress syndrome induced by RNA viruses including SARS-CoV-2 has been highlighted in the revised manuscript, Page 3, lines (76-79).

Thank you very much

Hope the revised manuscript finds your acceptance

Best Regards,

Rana Ahmed Youness, PhD

Assistant Professor of Molecular Biology and Biochemistry and Group Leader, Faculty of Biotechnology, German International University (GIU), New Administrative Capital, 11586, Cairo, Egypt. E-mail: [email protected]; [email protected]

Reviewer 4 Report

Comments and Suggestions for Authors

Rana A. Youness and collaborators presented a review focused on the role of hydrogen sulfide in oncological and non-oncological disorders and the regulation of its production by non-coding RNAs.

After a brief description of hydrogen sulfide function as a gasotransmitter and ncRNAs, the authors described the enzymes that generate hydrogen sulfide, and subsequently described the dysregulation of such enzymes in various tumors. Finally, the authors reported an in-depth review of cases of dysregulation of hydrogen sulfide production and/or expression of the genes producing it in pathological situations or different tissue contexts.

I have indications that authors should pay attention:

1) The main objection concerns the mechanism of action of H2S: The authors did not provide any description of the mode of action of H2S, a fundamental aspect of such review.

2) Figure 1 shows briefly how H2S is produced. Authors should provide in greater detail the chemical reactions catalyzed by each enzyme, either in the same figure or in a separate figure.

3) In table 2, it is necessary to add a further column indicating the tumor tissues or cell lines corresponding to the ncRNAs here reported.

4) Figure 4 and Table 2 appear to report a series of overlapping information. How do they provide differential information, and why do some data match, while others are absent in one or the other format?

5) Some ncRNAs reported in the table/figure have not been commented on in any paragraph of the text.

6) There is a paragraph missing regarding the conclusions.

7) Please check the subscripts of the chemical formulas.

Author Response

Kindly find herein a revised version of our manuscript, entitled “Role of Hydrogen Sulfide in Oncological and Non-Oncological Disorders and its Regulation by Non-coding RNAs: A Comprehensive Review”. Manuscript ID is ncRNA-2778534.

We are gratified for the opportunity to submit a revised version of our manuscript to your journal. In reply to the valuable comments of the reviewers and in the provision of the revised manuscript, I am sending you this letter providing the detailed changes that have been entailed in the revised manuscript (in red) for your kind consideration.

Dear Reviewers,

We are thankful for your thorough review of our manuscript that aims for the improvement and scientific enrichment of our paper. All the issues noted by reviewers have been taken into consideration and extensively revised. The comments and inquiries are shown in Bold and the replies are shown in red.

Reviewer 3:
Rana A. Youness and collaborators presented a review focused on the role of hydrogen sulfide in oncological and non-oncological disorders and the regulation of its production by non-coding RNAs.
After a brief description of hydrogen sulfide function as a gasotransmitter and ncRNAs, the authors described the enzymes that generate hydrogen sulfide, and subsequently described the dysregulation of such enzymes in various tumors. Finally, the authors reported an in-depth review of cases of dysregulation of hydrogen sulfide production and/or expression of the genes producing it in pathological situations or different tissue contexts.
I have indications that authors should pay attention:
1) The main objection concerns the mechanism of action of H2S: The authors did not provide any description of the mode of action of H2S, a fundamental aspect of such review.
According to the reviewer’s recommendation, a new section highlighting the mechanism of action of H2S in different physiological and pathological conditions have been added to the manuscript, Pages 12-13, lines (243-292).

2) Figure 1 shows briefly how H2S is produced. Authors should provide in greater detail the chemical reactions catalyzed by each enzyme, either in the same figure or in a separate figure.
According to the reviewer’s recommendation, all chemical reactions catalysed by each enzyme have been added in the revised manuscript Pages 6-7, lines (169-174).

.
3) In table 2, it is necessary to add a further column indicating the tumor tissues or cell lines corresponding to the ncRNAs here reported.
According to the reviewer’s recommendation, a new column have been added to Table 2 indicating the cell lines used in each study.
4) Figure 4 and Table 2 appear to report a series of overlapping information. How do they provide differential information, and why do some data match, while others are absent in one or the other format?
According to the reviewer’s recommendation, a meticulous revision have been done in this section so that all ncRNAs mentioned in text are present with all details in Table 2 and Figure 4 and we took this opportunity to add more very recently reported ncRNAs that were validated to directly affect H2S synthesizing enzymes.
5) Some ncRNAs reported in the table/figure have not been commented on in any paragraph of the text.
We are extremely grateful for the meticulous comment from the reviewer, Table 2 and Figure 4 are mainly dedicated to miRNAs and lncRNAs that are experimentally validated to directly target any of the H2S synthesizing enzymes. However, miR-640, miR-34a and miR-245-5p mentioned in the review are miRNAs that are reported to be altered upon exogenous H2S administration but are not validated regulators for H2S synthesizing enzymes. However, all validated ncRNAs that are directly targeting either CBS or CSE or 3-MST are thoroughly revised and very recent miRNAs from literature were included in text, table and figures such as miR-193 that was recently reported to target 3-MST.
6) There is a paragraph missing regarding the conclusions.
According to the reviewer’s recommendation, a conclusion and future recommendation sections have been added in the revised manuscript attached Pages 26-27, lines (618-637).
7) Please check the subscripts of the chemical formulas.

We are extremely grateful for the meticulous revision of our manuscript from the reviewer, all subscripts in the submitted revised manuscript has been revised and updated.

Thank you very much

Hope the revised manuscript finds your acceptance

Best Regards,

Rana Ahmed Youness, PhD

Assistant Professor of Molecular Biology and Biochemistry and Group Leader, Faculty of Biotechnology, German International University (GIU), New Administrative Capital, 11586, Cairo, Egypt. E-mail: [email protected]; [email protected]

Round 2

Reviewer 2 Report

Comments and Suggestions for Authors

To complete Figure 1, add the proposed regulatory roles for tRNA fragments under the tRNA fragment drawing (see  

Keam SP, Hutvagner G.Life (Basel). 2015, section:

3.3. Regulating Translational Efficiency

Aside from a role for tRFs in mediating miRNA/siRNA-like silencing, several studies suggest a role for tRFs in global translational repression. 

Also see:

Annu Rev Genet. 2020 Nov 23; 54: 47–69.

" In this review, we will discuss the non-canonical functions of tRNAs. These include tRNAs as precursors to novel small RNA molecules derived from tRNAs, also called tRNA-derived fragments, that are abundant across species and have diverse functions in different biological processes, including regulating protein translation, Argonaute-dependent gene silencing, and more."

Otherwise the reader will not know why the tRNA fragments were added.

Author Response

Kindly find herein a revised version of our manuscript, entitled “Role of Hydrogen Sulfide in Oncological and Non-Oncological Disorders and its Regulation by Non-coding RNAs: A Compre-hensive Review”. Manuscript ID is ncRNA-2778534.

We are gratified for the opportunity to submit a revised version of our manuscript to your journal. In reply to the valuable comments of the reviewers and in the provision of the revised manuscript, I am sending you this letter providing the detailed changes that have been entailed in the revised manuscript for your kind consideration.

Dear Reviewers,

We are thankful for your thorough review of our manuscript which aims for the improvement and scientific enrichment of our paper. All the issues noted by reviewers have been taken into consideration and extensively revised. The comments and inquiries are shown in Bold and the replies are shown in red.

Reviewer 2:

To complete Figure 1, add the proposed regulatory roles for tRNA fragments under the tRNA fragment drawing (see 

Keam SP, Hutvagner G.Life (Basel). 2015, section:

3.3. Regulating Translational Efficiency

Aside from a role for tRFs in mediating miRNA/siRNA-like silencing, several studies suggest a role for tRFs in global translational repression.

Also see:

Annu Rev Genet. 2020 Nov 23; 54: 47–69.

" In this review, we will discuss the non-canonical functions of tRNAs. These include tRNAs as precursors to novel small RNA molecules derived from tRNAs, also called tRNA-derived fragments, that are abundant across species and have diverse functions in different biological processes, including regulating protein translation, Argonaute-dependent gene silencing, and more.

Otherwise the reader will not know why the tRNA fragments were added.

We are extremely thankful for the critical eye of our reviewer, we have updated figure 1 to add the sentence that tRNA has been recently reported to act as a precursor for novel small RNA molecules known as tRNA fragments and also the roles of the tRNA fragments are listed in a text box below tRNA fragments in the figure 1 based on the reviewer’s recommendations

Thank you very much

Hope we the submitted revised manuscript receive your acceptance

Best Regards,

Rana Ahmed Youness, PhD

Assistant Professor of Molecular Biology and Biochemistry and Group Leader, Faculty of Biotechnology, German International University (GIU), New Administrative Capital, 11586, Cairo, Egypt. E-mail: [email protected]; [email protected]

Rana Ahmed Youness, PhD

Assistant Professor of Molecular Biology and Biochemistry and Group Leader, Faculty of Biotechnology, German International University (GIU), New Administrative Capital, 11586, Cairo, Egypt. E-mail: [email protected]; [email protected]

Reviewer 4 Report

Comments and Suggestions for Authors

The authors adequately responded to all my objections. 

Author Response

Kindly find herein a revised version of our manuscript, entitled “Role of Hydrogen Sulfide in Oncological and Non-Oncological Disorders and its Regulation by Non-coding RNAs: A Compre-hensive Review”. Manuscript ID is ncRNA-2778534.

We are gratified for the opportunity to submit a revised version of our manuscript to your journal. In reply to the valuable comments of the reviewers and in the provision of the revised manuscript, I am sending you this letter providing the detailed changes that have been entailed in the revised manuscript for your kind consideration.

Dear Reviewers,

We are thankful for your thorough review of our manuscript which aims for the improvement and scientific enrichment of our paper. All the issues noted by reviewers have been taken into consideration and extensively revised. The comments and inquiries are shown in Bold and the replies are shown in red.

Reviewer 4:

The authors adequately responded to all my objections.

We are extremely thankful for the reviewer’s positive feedback about implementing all his/her comments that positively impacted our manuscript quality and scientific content

Thank you very much

Hope we always meet peer-reviewer’s expectations

Rana Ahmed Youness, PhD

Assistant Professor of Molecular Biology and Biochemistry and Group Leader, Faculty of Biotechnology, German International University (GIU), New Administrative Capital, 11586, Cairo, Egypt. E-mail: [email protected]; [email protected]